# Lithium Chloride Protects against Sepsis-Induced Skeletal Muscle Atrophy and Cancer Cachexia

**DOI:** 10.3390/cells10051017

**Published:** 2021-04-26

**Authors:** Ji-Hyung Lee, Seon-Wook Kim, Jun-Hyeong Kim, Hyun-Jun Kim, JungIn Um, Da-Woon Jung, Darren R. Williams

**Affiliations:** New Drug Targets Laboratory, School of Life Sciences, Gwangju Institute of Science and Technology, 1 Oryong-Dong, Buk-Gu, Gwangju 61005, Korea; jihyung@gm.gist.ac.kr (J.-H.L.); sunwook9415@gist.ac.kr (S.-W.K.); kjhuioph@gist.ac.kr (J.-H.K.); khjun9013@gist.ac.kr (H.-J.K.); kajamium@gmail.com (J.U.)

**Keywords:** cancer cachexia, intensive care unit-acquired weakness, skeletal muscle wasting, lithium chloride, glycogen synthase kinase-3β, sepsis

## Abstract

Inflammation-mediated skeletal muscle wasting occurs in patients with sepsis and cancer cachexia. Both conditions severely affect patient morbidity and mortality. Lithium chloride has previously been shown to enhance myogenesis and prevent certain forms of muscular dystrophy. However, to our knowledge, the effect of lithium chloride treatment on sepsis-induced muscle atrophy and cancer cachexia has not yet been investigated. In this study, we aimed to examine the effects of lithium chloride using in vitro and in vivo models of cancer cachexia and sepsis. Lithium chloride prevented wasting in myotubes cultured with cancer cell-conditioned media, maintained the expression of the muscle fiber contractile protein, myosin heavy chain 2, and inhibited the upregulation of the E3 ubiquitin ligase, Atrogin-1. In addition, it inhibited the upregulation of inflammation-associated cytokines in macrophages treated with lipopolysaccharide. In the animal model of sepsis, lithium chloride treatment improved body weight, increased muscle mass, preserved the survival of larger fibers, and decreased the expression of muscle-wasting effector genes. In a model of cancer cachexia, lithium chloride increased muscle mass, enhanced muscle strength, and increased fiber cross-sectional area, with no significant effect on tumor mass. These results indicate that lithium chloride exerts therapeutic effects on inflammation-mediated skeletal muscle wasting, such as sepsis-induced muscle atrophy and cancer cachexia.

## 1. Introduction

Inflammation-associated skeletal muscle wasting involves the activation of the transcriptional complex nuclear factor kappa-light-chain-enhancer of activated B cells (NF-κB) via increase in circulating cytokines, such as interleukin-1β (IL-1β) [1]. Cancer cachexia and intensive care unit-acquired weakness (ICUAW) produced by sepsis are two prominent examples of inflammation-associated skeletal muscle wasting [1,2,3]. Both conditions result from acute-phase responses and systemic inflammation [4,5]. Cancer cachexia affects approximately 50% of all cancer patients and can be an immediate cause of death by increasing the side effects of chemotherapy [6]. ICUAW occurs in approximately 90% of patients with severe sepsis and increases morbidity and mortality [7]. For patients with cancer cachexia, there are limited treatment options, and exercise programs are not always feasible because of issues such as chronic fatigue and anemia [8,9]. ICUAW places a huge financial burden on healthcare systems [5,10]. Therefore, the development of effective therapeutics for inflammation-associated muscle wasting is a research priority.

Lithium has been termed an “Oldie but Goodie” drug, as it was first used to treat psychiatric disorders in the nineteenth century [11,12]. It is a commonly prescribed treatment for bipolar disorder and is included in the World Health Organization’s list of essential medicines [13]. Lithium chloride (LiCl) is also widely used in basic research as a glycogen synthase kinase-3β (GSK-3β) inhibitor [14].

Lithium has been shown to modulate multiple signaling pathways in cells. For example, lithium may protect against oxidative stress via the upregulation of complex I, II and III of the electron transport chain in mitochondria [15,16]. Lithium also modulates dopamine coupled G-protein signalling by altering the function of G-protein subunits [15,17], and competes with magnesium for N-methyl-D-aspartate receptor (NMDA) glutamate receptor binding to increase the availability of glutamate in post-synaptic neurons [15,18]. In addition, lithium increases γ-aminobutyric acid (GABA) concentration in plasma and cerebral spinal fluid that produces neuroprotective effects [15,19]. Therapeutically relevant concentrations of lithium also inhibit inositol monophosphatase, which reduces the activity of the G-protein-coupled receptor linked to phospholipase C [20]. Other effects of lithium in cells include modulation of the cyclic AMP second messenger system and enhancement of autophagy by reduced sequestosome-1 and increased LC3II expression [15,21,22].

Owing to its multiple effects in cells, lithium has been studied as a drug repurposing candidate for numerous diseases, such as enterovirus infection, Parkinson’s disease, radiation injury, and rheumatoid arthritis [23,24,25,26]. LiCl treatment has been shown to enhance muscle differentiation in vitro [27]. In addition, LiCl has been reported to prevent muscle fatigue and correct weaknesses in a model of limb-girdle muscular dystrophy 1D [28,29]. However, to our knowledge, the effect of LiCl on sepsis-induced muscle protein turnover has only been validated in vitro [30], and there are no published research papers on lithium treatment and cancer cachexia. In 2008, Sonni et al. [31] published a meeting abstract investigating the effect of lithium on muscle weight in cancer, but no follow-up paper has been published in the PubMed database.

In this study, we investigated the effects of LiCl on these two forms of inflammation-associated skeletal muscle wasting using in vitro and in vivo models of cancer cachexia and sepsis.

## 2. Materials and Methods

### 2.1. Reagents

Lipopolysaccharide (LPS) was purchased from Thermo Fisher Scientific, Waltham, USA. LiCl was purchased from Sigma-Aldrich (St. Louis, MO, USA). Ebselen was purchased from Tokyo Chemical Industry, Japan. Antibodies against myosin heavy chain 2 (MyHC; catalog number sc-53095) were purchased from Santa Cruz Biotechnology (Dallas, TX, USA), while Phospho (Ser-9) and naïve GSK3β (#9336 and #9315) antibodies were purchased from Cell Signaling Technology (Danvers, MA, USA). Alpha-tubulin (PA5-29444) antibody was purchased from Invitrogen (Waltham, MA, USA).

### 2.2. Cell Culture

C2C12 murine skeletal muscle precursor cells (myoblasts) were purchased from Koram Biotech Corp. (Seoul, Korea) and maintained in growth media (GM) consisting of Dulbecco’s modified Eagle’s medium (DMEM) supplemented with 10% fetal bovine serum (FBS), 50 units/mL penicillin, and 50 mg/mL streptomycin (PenStrep). Myoblasts were induced to differentiate into myotubes by treatment with differentiation media (DM: DMEM supplemented with 2% horse serum and PenStrep for 3 days). MyHC-positive nuclei were counted using Image J 1.52 software (National Institutes of Health, Bethesda, MD, USA). RAW 264.7 murine macrophages and CT-26 murine colon carcinoma cell line were purchased from the Korean Cell Line Bank (Seoul, Korea) and cultured in GM.

### 2.3. Collection of Conditioned Media

To prepare cancer cell-conditioned media (CCM), CT26 colon carcinoma cells were seeded in 100-mm culture plates at a density of 2 × 10^6^ cells/plate. After 48 h, the culture medium was changed to serum-free medium. CCM was harvested after 24 h, and debris was removed via centrifugation (1500 rpm for 3 min at 4 °C) and filtered with a 0.22-μm disk filter (Corning, New York, NY, USA).

### 2.4. MTT Assay for Cell Proliferation

The 3-(4,5-dimethylthiazol-2-yl)-2,5-diphenyltetrazolium bromide (MTT) assay was used to analyze cell proliferation as previously described [32]. Myoblasts were seeded in a 96-well plate at a density of 1.5 × 10^3^ cells per well. After stabilization for 24 h, the compound of interest and/or CCM was applied for an additional 48 h. The medium was then changed to MTT solution (0.5 mg/mL; final concentration) and incubated at 37 °C in a 5% CO_2_ incubator for 60 min. Dimethyl sulfoxide (DMSO; 50 μL) was added to solubilize, and the optical density was measured at 570 nm using a microplate reader (VersaMax, Molecular Devices, San Jose, CA, USA).

### 2.5. Flow cytometry

The cell death rate was measured by Annexin V and Propidium Iodide (PI) staining (BD Pharmingen, San Diego, CA, USA) following the manufacturer’s instructions. Briefly, C2C12 myoblasts were treated for 48 h with a designated concentration of LiCl; myoblasts treated with staurosphorine (Sigma-Aldrich) for 3 h were used as a positive control. Cell staining was performed for 15 min at room temperature and analyzed using FACSCanto II (BD Bioscience, Franklin Lakes, NJ, USA) within 1 h. Data analysis was conducted using FlowJo v10.7 software (BD Bioscience, Franklin Lakes, NJ, USA).

### 2.6. siRNA-Mediated Gene Knockdown

Genetic knockdown of Inositol-phosphate phosphatase (IMPA) was performed using Lipofectamine 3000 reagent according to the manufacturer’s guidelines (Thermo Fischer Scientific, Waltham, MA, USA). The differentiated myotubes were treated with CCM with control siRNA or siIMPA for 72 h. The transfected cells were fixed in 4% formaldehyde or lysed for further experiments.

### 2.7. Real-Time Quantitative PCR

The RNAs were extracted using TRIzol reagent following the manufacturer’s instructions (Thermo Fisher Scientific) and reverse transcribed using AccuPower^®^ RT PreMix (Bioneer, Daejeon, Korea). Real-time PCR was performed with specific primers using 2× Power SYBR^®^ Green PCR Master Mix (Waltham, MA, USA). For amplification and detection, the StepOnePlus Real-Time PCR System (Applied Biosystems) was used according to the manufacturer’s instructions. The expression levels of all genes (as indicated in the text) were normalized to the expression level of *Gapdh*. The primer details are presented in Table 1.

### 2.8. In Vitro Model of Cancer Cachexia and Morphological Analysis of Myotubes

Myoblasts were seeded in a 12-well plate at a density of 3 × 10^5^ cells/well and stabilized for 24 h in GM. The medium was changed to DM, and myoblasts were incubated in DM for 72 h to produce myotubes. The myotubes were then treated with a 1:1 ratio of DM and CCM for 72 h to induce myotube wasting. Compounds of interest were added to the CCM mixture before administration. Next, the myotubes were fixed in 4% formaldehyde and permeabilized with phosphate-buffered saline (PBS) containing 0.5% Triton X-100 (Sigma-Aldrich). MyHC antibody (Santa Cruz) was incubated in 1% bovine serum albumin in PBST (0.02% Tween 20 in PBS) overnight at 4 °C. The myotubes were incubated with the secondary antibody, Alexa Fluor 488 (Thermo Fisher), for 1 h at room temperature. Fluorescence images were taken in five different areas using a DMI 3000 B (Leica Microsystems GmbH, Wetzlar, Germany) and analyzed using Image J 1.52 software. MyHC-positive myotubes containing more than three nuclei were considered as myotubes, and their diameters were measured.

### 2.9. Immunoblotting

Immunoblotting was performed as previously described [33]. Briefly, the concentration of protein lysate was quantified using Bradford reagent (Bio-Rad, Hercules, CA, USA). After electrophoresis, the separated proteins were transferred onto Polyvinylidene fluoride (PVDF) membranes, blocked with 5% non-fat powdered milk in TBST (0.02% Tween 20 in TBS), and subsequently incubated overnight at 4 °C with the primary antibody. The secondary antibody was used at a 1:10,000 dilution (for mouse: Abcam, cat. no. ab6789; for rabbit: Cell Signaling Technology, cat. No. #7074). Detection was performed using ImageQuant™ LAS 500 (GE Healthcare, Chicago, IL, USA) following the manufacturer’s instructions and quantified using Image J 1.52 software.

### 2.10. Assessment of Pro-Cachexia Cytokine Induction in Macrophages

RAW 264.7 macrophages were seeded in 6-well culture plates at a density of 5 × 10^5^ cells/well. After 24 h, the macrophages were pre-treated with the compound of interest for 1 h, followed by treatment with 200 ng/mL LPS and LiCl for 24 h. RNA was then isolated using TRIzol reagent, and cytokine gene expression was measured using qPCR, as described in 2.7.

### 2.11. Animal Studies

Animal experiments were carried out in accordance with the ethical guidelines established by the Animal Care and Use Committees (ACUC) of the Gwangju Institute of Science and Technology, Republic of Korea (study approval number: GIST-2019-042). The animals were supplied by Damool Science, Republic of Korea.

### 2.12. Animal Model of Sepsis-Induced Muscle Wasting

Male C57BL/6 mice (10 weeks old) were randomly assigned to the following treatment groups: (1) saline group (intraperitoneal (IP) injection of saline (vehicle) every 24 h for 4 days); (2) LiCl group (IP delivery of 40 mg/kg LiCl every 24 h for 4 days); and (3) Untreated group. On the fourth day, the saline and LiCl groups were treated with 1 mg/kg LPS via IP injection, and the untreated group received saline via IP. Eighteen hours after LPS injection, the mice were anesthetized using ketamine (22 mg/kg; Yuhan, Korea) and xylazine (10 mg/kg; Bayer, Korea) and sacrificed. Mouse limb muscles were dissected, weighed, and frozen at −80 °C.

### 2.13. Animal Model of Cancer Cachexia

Male BALB/c mice (10 weeks old) were randomly assigned to the following treatment groups: (1) 1 × 10^6^ CT26 colon cancer cells inoculated subcutaneously (SC) in the right flank and (2) an equal volume of saline inoculated SC in the right flank. Tumor growth was monitored twice weekly, and tumor volume was measured using a caliper and calculated using the following formula: V (volume) = (longitudinal × transverse^2^)/2. Ten days after tumor injection, LiCl was administered daily at a specific concentration for 21 days. Grip strength was measured using a BIO-GS3 strength meter (Bioseb, France). Each mouse was tested four times at 30 min intervals, and the maximum value force was used to represent the muscle force. Twenty-eight days after CT26 cell or saline injection, the mice were anesthetized and sacrificed for further analysis.

### 2.14. Histological Analysis

The quadriceps were fixed in 4% paraformaldehyde solution overnight at 4 °C and embedded in paraffin. Each muscle section (5 µm) was stained with hematoxylin and eosin (H&E) using a kit (Merck, Darmstadt, Germany). Images were obtained using light microscopy (DM 2500, Leica Microsystems GmbH, Wetzlar, Germany), and the cross-sectional area was measured using Image J 1.52 software (National Institutes of Health, Bethesda, MD, USA).

### 2.15. Statistics

Microsoft Excel 2016 (Redmond, DC, USA) was used to determine statistical significance between two samples using the Student’s *t*-test. For multiple comparisons, ANOVA was performed to determine statistical significance. Tukey’s post hoc comparison of the means was performed using Origin Pro 9.1 software (OriginLab, Northampton, MA, USA). Statistical significance was set at *p* < 0.05. Unless otherwise stated in the figure legends, experiments were carried out in triplicate, and the error bars represent standard deviations.

## 3. Results

### 3.1. LiCl Enhanced Myogenic Differentiation

Prior to evaluating the effect of LiCl on cancer cachexia, we verified the effect of LiCl on myogenic differentiation. LiCl treatment at concentrations greater than 10 mM significantly reduced the proliferation of C2C12 myoblasts (Figure 1A). An increase in cell death rate was observed in 20 mM LiCl-treated C2C12 myoblasts (Figure 1B,C). Therefore, concentrations of 5 mM or lower were used to determine the effect on myogenesis. C2C12 myoblasts were differentiated into myotubes using differentiation media (DM) for 72 h with increasing concentrations of LiCl. Afterwards, cells were visualized using immunofluorescence staining. The number of MyHC-positive nuclei, a myocyte marker [34], was increased by 5 mM LiCl treatment, indicating enhanced myogenic differentiation (Figure 1D,E). Accordingly, we observed a reduction in Pax-7, a myoblast marker [34] and an increase in MyHC in the 24-h LiCl-treated group, confirming its pro-myogenic effect (Figure 1F, Appendix A). To validate the action of lithium, we measured the inhibitory phosphorylation (Ser9) of GSK3β [35] and found a significant increase in GSK3β phosphorylation at a concentration of 5 mM after 24 h of treatment (Figure 1E,F).

### 3.2. LiCl Increased Myh2 Expression and Reduced Pax-7 Expression in Differentiating Myoblasts Treated with CCM

To examine the effect of LiCl on proliferation and differentiation, C2C12 myoblasts were treated with CCM derived from CT26 murine colorectal cancer cells. As expected, CCM treatment reduced the number of MyHC-positive nuclei, indicating repression of myogenic capacity (Figure 2A,B). This impairment in myogenic potential was partially recovered by LiCl treatment (Figure 2A,B). CCM treatment decreased the protein levels of MyHC; however, LiCl increased the protein expression of MyHC in CCM-treated myoblasts (Figure 2D,E). Expression analysis of the markers *Pax-7* (myoblasts) and *Myh2* (myotubes) indicated that the impairment in myogenic capacity following CCM exposure was partially recovered by LiCl treatment (Figure 2F). To examine whether LiCl can ameliorate the CCM-induced decrease in myoblast proliferation, we performed an MTT proliferation assay. LiCl did not prevent the inhibition of myoblast proliferation (Appendix A).

### 3.3. LiCl Prevented CCM-Induced Myotube Wasting

Differentiated C2C12 myotubes were treated with CCM with or without LiCl. CCM-treated myotubes showed approximately 30% decrease in diameter, which was inhibited by co-treatment with LiCl (Figure 3A,B). Measurement of myotube diameter distribution indicated that LiCl preserved larger myotubes (Appendix A). LiCl prevented both *Pax-7* upregulation and MyHC downregulation in myotubes similar to that in differentiated myoblasts (Figure 3C–E). The assessment of Ser-9 phosphorylation of GSK3β indicated that LiCl treatment increased the inhibitory phosphorylation of GSK3β (Figure 3D,E). The E3 ubiquitin ligase, Atrogin-1 (MAFbx/FBXO32), is a major effector of increased muscle proteolysis in cancer cachexia [36,37]. Atrogin-1 expression was upregulated in myotubes treated with CCM and inhibited by treatment with LiCl (Figure 3C–E).

### 3.4. Lithium Mimetic Ebselen Did Not Prevent Myotube Wasting Induced by CCM

Lithium is also an inositol phosphate phosphatase (IMPase) inhibitor [20]. The inhibitor drug ebselen was used to investigate whether the effects of lithium on CCM-treated myotubes are influenced by IMPase inhibition. As in the previous experiment, CCM containing ebselen was treated to differentiating myoblasts and differentiated myotubes, respectively. Gene expression analysis showed that ebselen did not prevent the upregulation of *Atrogin-1* and *Pax-7* expression or the downregulation of *Myh2* induced by CCM (Appendix A). In addition, by using immunofluorescence staining of MyHC, we observed that ebselen did not attenuate the reduction in myotube diameter after CCM treatment (Figure 4A,B). A siRNA-mediated knockdown experiment was performed to confirm whether the effects of LiCl on CCM-induced myotube atrophy is independent of IMPA. Consistent with the ebselen experiment, genetic knockdown of IMPA did not attenuate CCM-induced myotube atrophy (Figure 4C–E). These results suggest that the protective effect of lithium on CCM-induced myotube atrophy is independent of IMPA inhibition.

### 3.5. LiCl Inhibited LPS-Induced Inflammatory Cytokine Production

It has been well recognized that pro-inflammatory cytokines, such as IL-1β, IL-6 and iNOS, are induced by LPS and can produce skeletal muscle atrophy [38,39]. LPS-treated macrophages have been used as an in vitro model of pro-inflammatory cytokine production during sepsis [38,39]. RAW264.7 macrophages were treated with 200 ng/mL LPS in the presence or absence of 5 mM LiCl, and the induction of the cachexia-related cytokines *Il-1b* [40], *Il-6* [41], and *inos* [42] was measured by qPCR. It was observed that LiCl treatment significantly reduced the induction of Il-1β, Il-6 and inos by LPS (Figure 5).

### 3.6. LiCl Prevented Muscle Wasting in a Mouse Model of Septic Cachexia

To investigate the protective effect of LiCl on inflammation-induced muscle wasting, we first established a mouse model of septic cachexia using LPS [43]. It has been reported that at least 200 mg/kg of lithium is needed to achieve a therapeutic dose for treating bipolar disorder in a mouse model [44]. However, other studies found that 40 mg/kg of lithium showed beneficial effects on the kidney and against irradiation-induced damage [45,46,47]. Kidney toxicity is known to be a major concern of lithium therapy [11], therefore we used 40 mg/kg for our experiments to minimize this issue, since it has been reported as a non-toxic dose for the kidney [45,46]. At 18 h post-LPS injection, there was a significant reduction in body weight compared to that in mice receiving saline treatments (Figure 6A). The administration of LiCl attenuated this decline in body weight (Figure 6A). The weight of quadriceps was measured to determine the effect of LiCl on muscle wasting. LPS treatment decreased muscle mass and attenuated LPS-induced muscle loss (Figure 6B). The effect of LiCl on sepsis-induced fiber wasting was assessed by measuring the cross-sectional area of quadriceps. LPS produced a significant decline in muscle fiber cross-sectional area, which was ameliorated by LiCl treatment (Figure 6C,D). In addition, the proportion of relatively large fibers was maintained in the LiCl-treated group (Appendix A). To evaluate muscle functionality, we performed a grip strength test. LPS administration significantly weakened grip strength, while LiCl treatment prevented the loss of muscle strength (Figure 6E). Skeletal muscle E3 ligases are known to mediate LPS-induced muscle wasting [43]. As expected, LPS treatment upregulated the expression of the E3 ligases *Atrogin-1* and *Murf-1*, while LiCl inhibited the expression of *Atrogin-1* and *Murf-1* (Figure 6F). As *inos* is known as an important regulator of cytokine-mediated muscle wasting [48,49], the expression of this gene was also measured. LiCl completely inhibited *iNOS* gene induction post-LPS administration (Figure 6F).

### 3.7. LiCl Treatment Attenuated Muscle Wasting in Cancer Cachexia

To further validate the anti-atrophic effect of LiCl on inflammation-induced muscle wasting, we used a syngeneic mouse model of cancer cachexia based on the transplantation of CT26 colon carcinoma cells into BALB/c mice, as previously described [50]. LiCl treatment did not significantly affect the overall body weight of CT26 tumor-bearing mice (Figure 7B and Appendix A). In addition, LiCl did not alter tumor growth (Figure 7A and Appendix A). Skeletal muscle and adipose tissue are among the most affected tissues in cancer cachexia [51]. Tumor-bearing mice showed a significant reduction in the weight of quadriceps muscle, tibialis anterior muscle, and gonadal adipose tissues. LiCl treatment significantly ameliorated quadriceps and tibialis anterior muscle loss (Figure 7C). In contrast, adipose tissue mass was not affected by LiCl (Figure 7D). Grip strength was measured to assess whether the improvement in muscle mass after LiCl treatment influenced muscle function. LiCl increased muscle function at the 80 mg/kg dose (Figure 7E). In line with the grip strength data, the 80 mg/kg dose preserved the average fiber cross-sectional area, with a higher proportion of fibers in the 4000–5000 µm^2^ range (Figure 7F,G and Appendix A). The increased expression of *Atrogin-1* and *Murf-1* in tumor-bearing mice was attenuated in both the 40 and 80 mg/kg LiCl-treated groups (Figure 7H). Besides E3 ligases, previous studies have demonstrated the role of IL-6 in the progression of cancer-mediated muscle wasting [52,53]. Significant upregulation of *Il-6* was observed in the skeletal muscle tissue of tumor-bearing mice, and the administration of LiCl completely inhibited the tumor-induced increase in *Il-6* expression (Figure 7H). Taken together, lithium administration increased muscle mass and enhanced muscle strength without affecting tumor progression and adipose tissue mass, indicating that lithium can be used as a therapeutic agent to manage cancer-induced loss of skeletal muscle.

## 4. Discussion

Inflammation-associated skeletal muscle wasting occurs in cancer cachexia and sepsis-induced skeletal muscle atrophy and significantly affects patient morbidity and mortality. Due to a lack of effective treatment options [54,55], this study aimed to investigate the potential of LiCl in treating inflammation-associated skeletal muscle wasting using in vitro and in vivo models of cancer cachexia and sepsis. We found that LiCl increased muscle mass, enhanced muscle strength, and increased fiber cross-sectional area.

LiCl reportedly induces myogenic differentiation in C2C12 myoblasts [56,57]. LiCl was effective in inhibiting wasting in both cell-based and animal models of cancer cachexia and sepsis-induced skeletal muscle atrophy. Animal studies revealed a difference in the LiCl dosage required for achieving therapeutic effects in LPS-induced muscle wasting and cancer cachexia (Figure 6 and Figure 7). The 40 mg/kg dose of LiCl was effective in the LPS model, whereas 80 mg/kg was required to achieve effectiveness in the cancer cachexia model. We selected the 80 mg/kg dose because it was lower than reported doses of LiCl that produced anti-cancer effects [58], whereas the aim of our study was to investigate cachexia related cachexia. The difference in the effective doses between the cancer cachexia and sepsis models may be due to differences in the treatment regimes [49]. In the cancer cachexia model, LiCl was treated 10 d after the transplantation of cancer cells. At this time point the inflammatory environment is already established. Thus, a higher dosage of LiCl would be needed to inhibit the progression of wasting in this model.

LiCl was effective at preserving muscle mass and function in tumor-bearing mice. However, the gonadal adipose tissue was unaffected by LiCl treatment. GSK-3β activity is known to increase in animal models with greater adiposity [59]. In addition, repression of Wnt signaling has been shown to enhance adipogenesis [60]. Therefore, LiCl-mediated GSK-3β inhibition and increased Wnt signaling may explain the lack of effect on adipose tissue mass. It should also be noted that skeletal muscle wasting rather than adipose tissue wasting is used in the formal definition of cancer cachexia [61]. Thus, treatments for cancer cachexia should ideally target skeletal muscle mass rather than adipose tissue.

Previous research has shown that lithium modulates multiple cell signaling pathways and processes [15,16,17,18,19,20,21,22]. At therapeutically relevant concentrations, lithium was demonstrated to inhibit the activities of glycogen synthase kinase-3β and inositol monophosphatase [62,63]. Our results indicate that glycogen synthase kinase-3β inhibition is the target of mechanism of LiCl in this study, because gene knockdown or chemical in-hibition of inositol monophosphatase using ebselen could not recapitulate the effects of LiCl on CM-induced myotube atrophy (Figure 4).

Lithium has been used as a neuroprotective agent in patients with cancer [64]. The protective effects of lithium on motor neurons may have contributed to the reduced muscle wasting observed in the current study. Our in vitro data indicated that LiCl directly prevents muscle wasting, because the treated myotubes showed reduced wasting and reduced expression of Atrogin-1 in the presence of CCM. CCM contains factors that inhibit both myoblast proliferation and myogenic differentiation [39]. The in vitro and in vivo experiments in this study utilized the CT26 colon carcinoma cell line because cancer cachexia is commonly associated with this tumor type [65]. Cancer cachexia is a common complication of lung, breast, and pancreatic carcinoma [66]. In future, it may be interesting to examine the effectiveness of LiCl in treating cachexia in models of these tumor types.

Notably, the effect of LiCl was rapid in this study (96 h treatment for the sepsis model). However, lithium takes longer to become effective in patients with bipolar disorder, usually around 6–8 weeks [11]. Although long-term lithium use has been reported to be associated with kidney toxicity [11], subsequent research has indicated the need to re-evaluate the effect of lithium on kidney function, owing to reports of protective outcomes in models of acute and chronic kidney disease [67]. Despite the association of Wnt pathway activation with cancer progression, long-term lithium therapy has not been shown to increase cancer risk [68]. Moreover, the potentially beneficial effects of lithium therapy were revealed in analyses showing enhanced life expectancy in humans and metazoans [69]. These reports support the potential development of lithium therapy for patients with muscle wasting.

In summary, the results presented herein show that LiCl treatment is effective in two animal models of inflammation-mediated skeletal muscle wasting, namely, cancer cachexia induced by tumor-derived factors and endotoxin-induced muscle weakness resulting from sepsis. A two-fold higher dose was required for effectiveness in the cancer cachexia model than in the sepsis model. Further studies are warranted to assess the potential of LiCl treatment in other types of skeletal muscle wasting, such as aging-related sarcopenia, sarcopenic obesity, and immobilization-induced muscle wasting.

## Figures and Tables

**Figure 1 cells-10-01017-f001:**
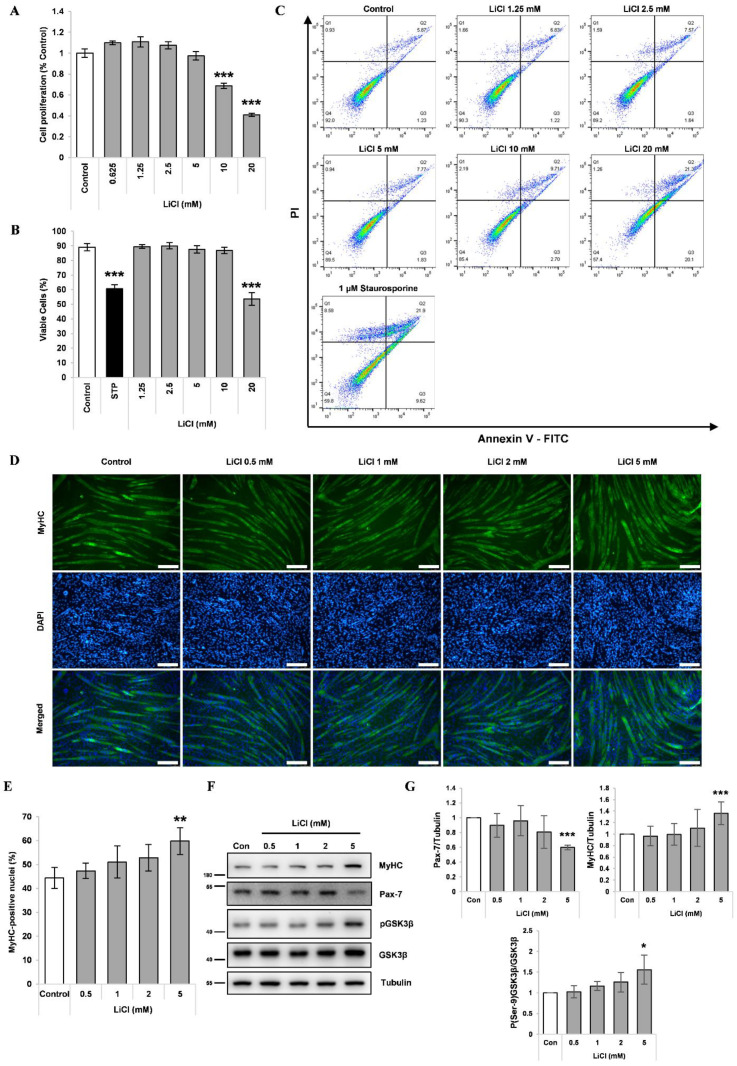
LiCl enhances myogenic differentiation in C2C12 myoblasts. (**A**) MTT proliferation assay of C2C12 myoblasts after 48 h of treatment with the designated concentration of LiCl (n = 5). (**B**) Quantification results of cell death rates in LiCl-treated C2C12 myoblasts (n = 4). (**C**) Annexin V and PI staining results from C2C12 cells treated with the indicated concentrations of LiCl and staurosphorine. (**D**) Representative immunofluorescence staining images targeting MyHC after 72 h of differentiation following treatment with increasing concentrations of LiCl. Scale bar represents 100 µm. (**E**) Measured ratio of MyHC-positive nuclei in immunofluorescence staining (n = 5). (**F**) Representative images of Western blotting for MyHC, Pax-7, phospho (Ser-9) GSK3β, naïve GSK3β, and alpha-Tubulin after 24 h treatment in differentiation media (DM). (**G**) Quantification results of Western blotting (n = 4). Statistical significance compared to the control is indicated by * (* = *p*-value < 0.05, ** = *p*-value < 0.01, *** = *p*-value < 0.001).

**Figure 2 cells-10-01017-f002:**
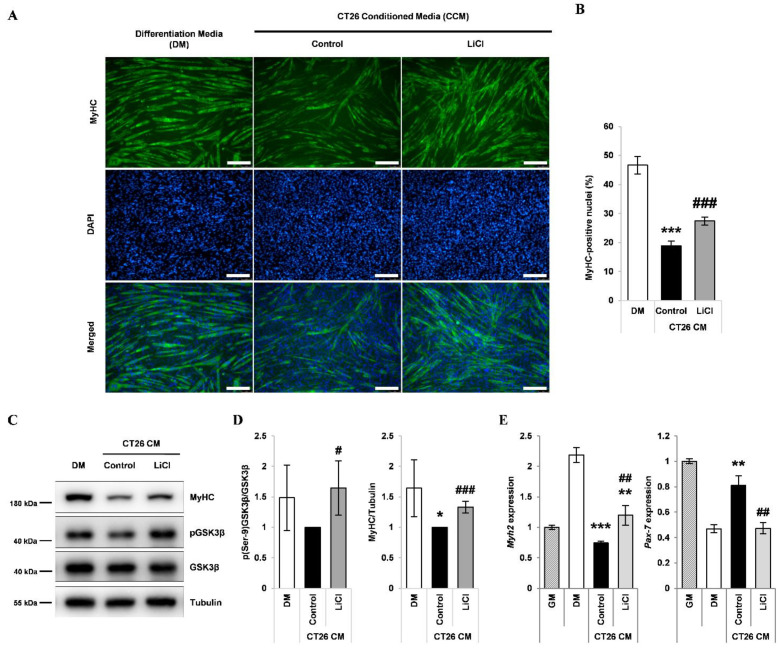
LiCl ameliorates the inhibition of myoblast differentiation by CT26 colon carcinoma cell-conditioned media (CCM). (**A**) Immunofluorescence staining after 72 h of culture in the following media: (1) differentiation media (DM), (2) CCM only, and (3) CCM and 5 mM LiCl (n = 4). (**B**) Quantified ratio of MyHC-positive nuclei in immunostaining images (n = 4). (**C**) Representative Western blot images for myosin heavy chain (MyHC), phospho (Ser-9) GSK3β, naïve GSK3β, and alpha-tubulin after 72 h of culture in (1) DM, (2) CCM, (3) CCM and 5 mM LiCl (n = 3). (**D**) Quantification results of Western blot analysis (n = 3). (**E**) qPCR analysis of the myotube marker, *Myh2*, and the myoblast marker, *Pax-7*, after 24 h of incubation in the following media: (1) GM, (2) DM, (3) CCM, (4) CCM and 5 mM LiCl (n = 5). Statistical significance compared to DM is indicated by * (* = *p*-value < 0.05, ** = *p*-value < 0.01, *** = *p*-value < 0.001). # represents statistical significance compared to the CCM group (# = *p*-value < 0.05, ## = *p*-value < 0.01, ### = *p*-value < 0.001).

**Figure 3 cells-10-01017-f003:**
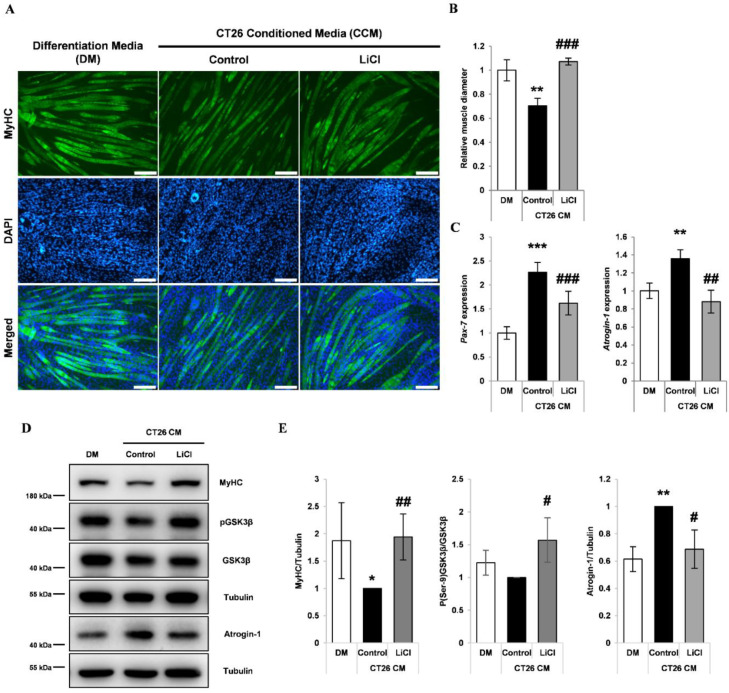
LiCl prevents CT26 colon carcinoma cell-conditioned media (CCM)-mediated wasting in C2C12 myotubes. (**A**) Immunofluorescence staining for MyHC after 72 h of culture in (1) differentiation media (DM), (2) CCM, and (3) CCM and 5 mM LiCl. DAPI staining was used to visualize cell nuclei. Scale bar = 100 µm (n = 4). (**B**) Quantification of relative myotube diameter (n = 4). (**C**) qPCR of *Pax-7* and *Atrogin-1* expression in myotubes cultured with (1) DM at the start of the experiment, (2) DM for 24 h, (3) CCM for 24 h, and (4) CCM and 5 mM LiCl for 24 h (n = 3). (**D**) Western blotting for MyHC, phospho (Ser-9) GSK3β, naïve GSK3β, and alpha-Tubulin in C2C12 myotubes after 72 h culture in (1) DM, (2) CCM, and (3) CCM and 5 mM LiCl (n = 4). Western blotting for Atrogin-1 and alpha-Tubulin (lower) in C2C12 myotubes after 24 h culture in (1) DM, (2) CCM, and (3) CCM and 5 mM LiCl (n = 3). (**E**) Quantification results of Western blotting for MyHC (n = 4), phospho (Ser-9) GSK3β (n = 3), and Atrogin-1 (n = 3). Significance compared to DM treatment is indicated by * (* = *p*-value < 0.05, ** = *p*-value < 0.01, *** = *p*-value < 0.001). Significance compared to CCM treatment is indicated by # (# = *p*-value < 0.05) ## = *p*-valve < 0.01, ### = *p*-value < 0.001).

**Figure 4 cells-10-01017-f004:**
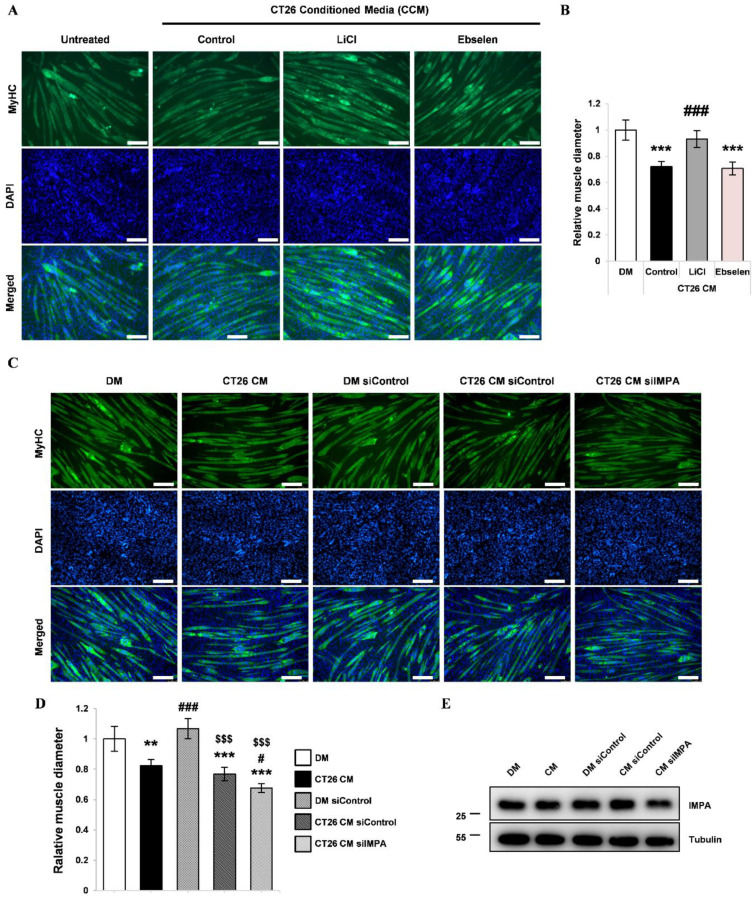
Effect of the lithium mimetic and IMPase inhibitor, ebselen, on CCM-induced wasting in C2C12 myotubes (**A**) Immunofluorescence staining for MyHC after 72 h of culture in (1) DM, (2) CCM, (3) CCM and 5 mM LiCl, and (4) CCM and 10 µM ebselen. DAPI staining was used to visualize cell nuclei. Scale bar = 100 µm. (n = 3). (**B**) Quantification of relative myotube diameter. (n = 3). (**C**) Representative immunofluorescence staining images for MyHC after 72 h of the designated treatments. (**D**) Quantification of relative myotube diameters (n = 4). (**E**) Representative Western blot images for IMPA (n = 4). Significance compared to DM is indicated by * (** = *p*-value < 0.01, *** = *p*-value < 0.001). Significance compared to CT26 CM (CCM) treatment is indicated by # (# = *p*-value < 0.05, ### = *p*-value < 0.001). Statistical significance relative to siControl-treated DM is indicated by $ ($$$ = *p*-value < 0.001).

**Figure 5 cells-10-01017-f005:**
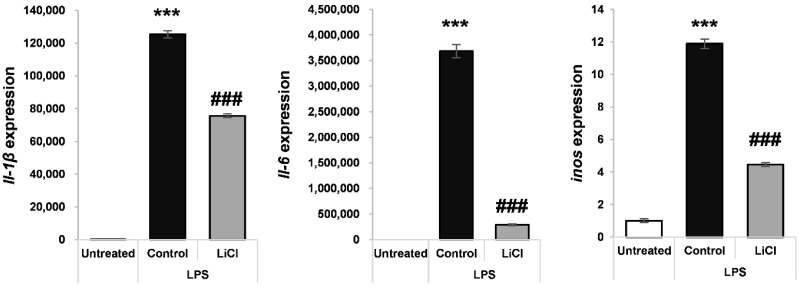
LiCl treatment blocks LPS-induced upregulation of inflammatory cytokines in RAW264.7 macrophages. qPCR analysis of *IL-1β*, *iNOS*, and *IL-6* expression after treatment with 5 mM LiCl treatment for 24 h, followed by 5 mM LiCl and 200 ng/mL LPS for an additional 24 h. Significance compared to no treatment is indicated by * (*** = *p*-value < 0.001). Significance compared to LPS treatment is indicated by # (### = *p*-value < 0.001).

**Figure 6 cells-10-01017-f006:**
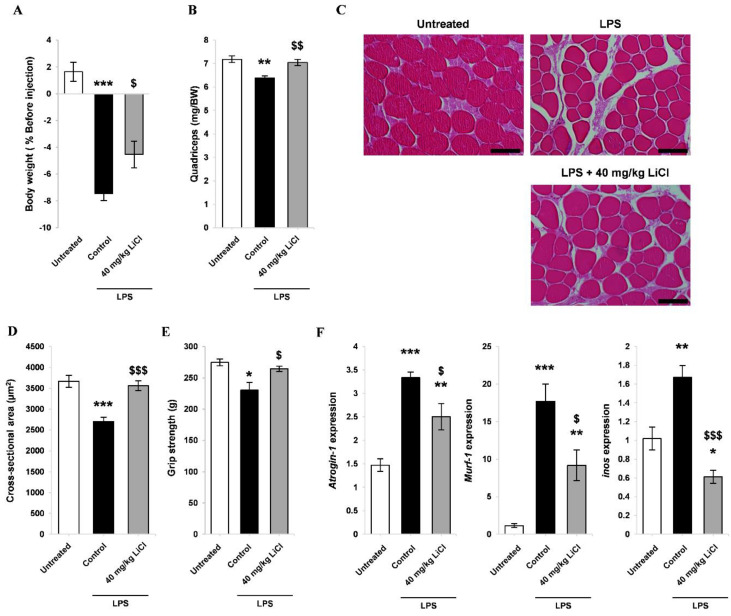
Therapeutic efficacy of LiCl on skeletal muscle wasting in a mouse model of sepsis. Male BALB/c mice were treated as follows: (1) no treatment, (2) saline for 96 h followed by 1 mg/kg LPS, and (3) 40 mg/kg LiCl for 96 h followed by 1 mg/kg LPS. (**A**) Relative body weight change after 18 h (n = 7). (**B**) Quadriceps weight/body weight (mg/BW) ratio (n = 6). (**C**) Representative H&E staining images of the quadriceps (scale bar = 100 µm). (**D**) Average fiber cross-sectional area of quadriceps (n = 5). (**E**) Grip strength test results after LPS treatment (n = 5). (**F**) qPCR analysis of *inos*, *Atrogin-1*, and *Murf-1* in the quadriceps (n = 5). Significant difference compared to untreated is indicated by * (* = *p*-value < 0.05, ** = *p*-value < 0.01, *** = *p*-value < 0.001). Significant difference compared to saline plus LPS (control) is indicated by$ ($ = *p*-value < 0.05, $$ = *p*-value < 0.01, $$$ = *p*-value < 0.001).

**Figure 7 cells-10-01017-f007:**
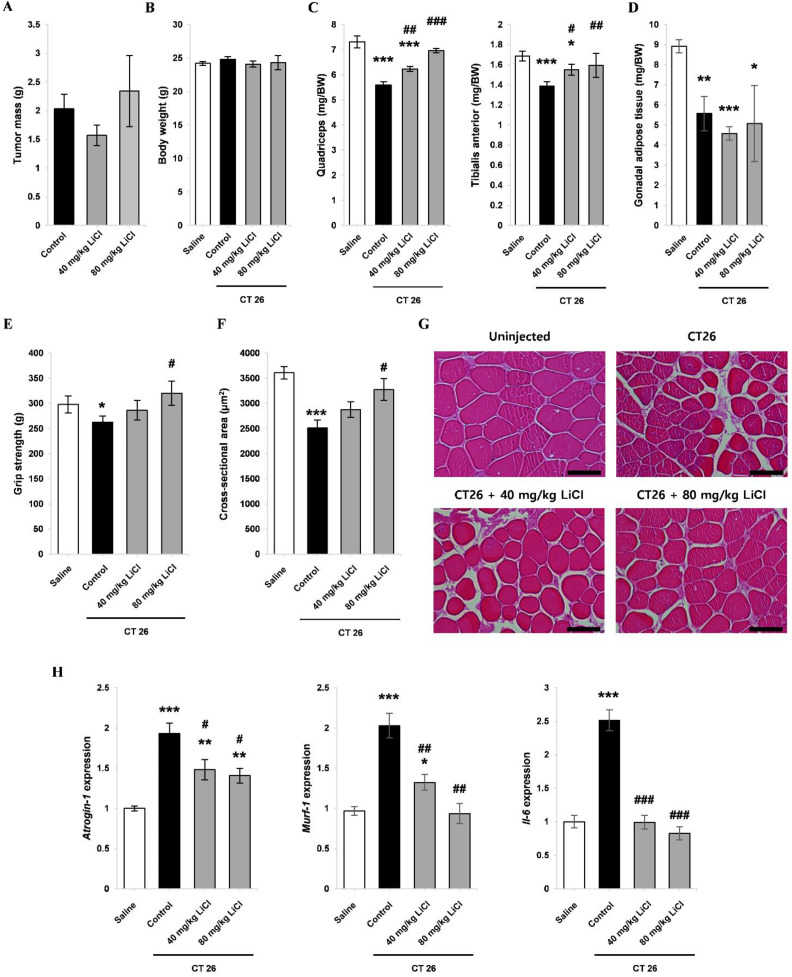
LiCl ameliorates muscle wasting in a mouse model of cancer cachexia. Male BALB/c mice were treated as follows: (1) no treatment, (2) transplantation with CT26 colon carcinoma cells, (3) transplantation with CT26 cells and treatment with 40 mg/kg LiCl, and (4) transplantation with CT26 cells and treatment with 80 mg/kg LiCl. (**A**) Dissected tumor mass. (**B**) Tumor free body weight. (**C**) Quadriceps and tibialis anterior muscle weight/body weight (mg/BW) ratio. (**D**) Gonadal adipose tissue mass for (**A**–**D**) n = 5 at 28 days post-transplantation. (**E**) Grip strength at 21 days post-transplantation (n = 5). (**F**) Average fiber cross-sectional area of quadriceps (n = 4). (**G**) Representative H&E staining images of the quadriceps (scale bar = 100 µm). (**H**) qPCR analysis of Atrogin-1, Murf-1, and Il-6 expression in quadriceps muscle (n = 4). Significant difference compared to saline is indicated by* (* = *p*-value < 0.05, ** = *p*-value < 0.01, *** = *p*-value < 0.001). Significant difference compared to CT26 control is indicated by# (# = *p*-value < 0.05, ## = *p*-value < 0.01, ### = *p*-value < 0.001).

**Table 1 cells-10-01017-t001:** qPCR primers used in this study.

Gene	Locus	Source		Primer Sequence	Size
*Myh2*	NM_001039545	*Mus musculus*	Forward	GAAGAGCCGGGAGGTTCAC	113 bp
			Reverse	ACACAGGCGCATGACCAAA	
*Pax-7*	NM_011039	*Mus musculus*	Forward	CCCTTTCAAAGACCAAATGCA	198 bp
			Reverse	CCCTCACGGGCAGATCATTA	
*Atrogin-1*	NM_026346	*Mus musculus*	Forward	CAGAGAGCTGCTCCGTCTCA	178 bp
			Reverse	ACGTATCCCCCGCAGTTTC	
*Murf-1*	NM_001039048	*Mus musculus*	Forward	CCGAGTGCAGACGATCATCTC	198 bp
			Reverse	TGGAGGATCAGAGCCTCGAT	
*Il-6*	NM_031168	*Mus musculus*	Forward	GAGGATACCACTCCCAAC	141 bp
			Reverse	AAGTGCATCATCGTTGTT	
*inos*	NM_010927	*Mus musculus*	Forward	CCCCTTCAATGGCTGGTACA	64 bp
			Reverse	GCGCTGGACGTCACAGAA	
*Il-1β*	NM_008361	*Mus musculus*	Forward	TGCCACCTTTTGACAGTGATG	135 bp
			Reverse	TGTGCTGCTGCGAGATTTGA	
*Gapdh*	NM_001289726	*Mus musculus*	Forward	CTCCACTCACGGCAAATTCA	120 bp
			Reverse	GCCTCACCCCATTTGATGTT	

## Data Availability

The data presented in this study are available in this article.

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
