# Peer review of "Lithium Chloride Protects against Sepsis-Induced Skeletal Muscle Atrophy and Cancer Cachexia"

_cells, 2021, doi:10.3390/cells10051017_

Round 1

Reviewer 1 Report

The manuscript is dedicated to the effects of LiCl on the development of some muscle diseases of systemic origin: intensive care unit acquired weakness (ICUAW) produced by sepsis and cancer cachexia. Both diseases are associated with the inflammatory process. The study uses the two main approaches:in vitro and and in vivo tests. In all models used the aplied LiCl dosage had the significant efficient results.  The study design is excellent, all the necessary tests were done and properly analyzed. 

The  only concern I have is not appropriate mechanistic considerations in the Introduction and Discussion. The authors insist that LiCl influences on the numerous intracellular mechanisms, but really mentioned the only one of them : Ser9 Phosphorylation of GSK3beta. As far as I know this mechanism is very important and can explain many effects described by the authors. Anyway it will be better if the authors add some words to the Introduction and Discussion regarding the other signalling pathways which could be subjected to LiCl influence. 

I consider the paper very good and after this minor revision ready for publication.

Reviewer 2 Report

The paper by Lee et al. reports the effect of lithium chloride in sepsis-induced skeletal muscle atrophy and cancer cachexia.

The study has been nicely performed and designed even if sometimes the huge quantities of results get the paper clear less. I would accept this work for publication but with minor revision:

- I suggest to review the language.

- on line 62 I suggest to delete the results.

- In the results, I think that there are too many figures and panels. I suggest leaving only the most significant and move the others to Supplementary Materials.

- Concerning 3.1 I suggest to better explain the results and clarify non only the concentration but also the chosen timing.

- In 3.4, 3.5 and 3.6 I suggest to better elucidate the experiment and the type of test performed. For instance, in 3.6 considering the wide range (200 mg/kg vs 40 mg/kg) explain what guided your choice.   

- In the discussion add references on line 386, 391, 394, and better explain the results obtained in this study respect the other taken from the literature: sometimes it is difficult understand if you are referring to your own or to other studies.
